# HPV Tests Comparison in the Detection and Follow-Up after Surgical Treatment of CIN2+ Lesions

**DOI:** 10.3390/diagnostics12102359

**Published:** 2022-09-29

**Authors:** Fabio Bottari, Anna Daniela Iacobone, Davide Radice, Eleonora Petra Preti, Mario Preti, Dorella Franchi, Sara Boveri, Maria Teresa Sandri, Rita Passerini

**Affiliations:** 1Division of Laboratory Medicine, European Institute of Oncology IRCCS, 20139 Milan, Italy; 2Department of Biomedical Sciences, University of Sassari, 07100 Sassari, Italy; 3Preventive Gynecology Unit, European Institute of Oncology IRCCS, 20139 Milan, Italy; 4Division of Epidemiology and Biostatistics, European Institute of Oncology IRCCS, 20139 Milan, Italy; 5Department of Surgical Sciences, University of Torino, 10124 Torino, Italy; 6Scientific Directorate, IRCCS Policlinico San Donato, 20097 San Donato, Italy; 7Bianalisi Laboratory, Carate Brianza, 20841 Carate Brianza, Italy

**Keywords:** HPV, cervical intraepithelial neoplasia, follow-up, real time PCR, genotyping, concordance

## Abstract

Background: HPV tests differ for technology, targets, and information on genotyping of high risk (HR) HPV. In this study, we evaluated the performance of 6 HPV DNA tests and one mRNA test in the detection of cervical intraepithelial lesions (CIN) and as a test-of-cure in the follow-up after surgical conservative treatment. Methods: One hundred seventy-two women referred to the European Institute of Oncology, Milan, for surgical treatment of pre-neoplastic cervical lesions, were enrolled in this study (IEO S544) from January 2011 to June 2015. For all women, a cervical sample was taken before treatment (baseline) and at the first follow-up visit (range 3 to 9 months): on these samples Qiagen Hybrid Capture 2 (HC2), Roche Linear Array HPV Test (Linear Array), Roche Cobas 4800 HPV test (Cobas), Abbott RealTime High Risk HPV test (RT), BD Onclarity HPV assay (Onclarity), Seegene Anyplex II HPV HR Detection (Anyplex), and Hologic Aptima HPV Assay (Aptima) histology and cytology were performed at baseline, and the same tests and cytology were performed at follow-up. Results: At baseline 158/172 (92%), histologies were CIN2+, and 150/172 (87%) women were recruited at follow-up. Assuming HC2 as a comparator, the concordance of HPV tests ranges from 91% to 95% at baseline and from 76% to 100% at follow-up (PABAK ranging from 0.81 to 0.90 at baseline and PABAK ranging from 0.53 to 1 at follow-up). All HPV showed a very good sensitivity in CIN2+ detection at baseline, more than 92%, and a very good specificity at follow-up, more than 89%. Conclusions: HPV tests showed a good concordance with HC2 and a very good and comparable sensitivity in CIN2+ detection. Hence, an HPV test represents a valid option as test-of-cure in order to monitor patients treated for CIN2+ lesions during follow-up.

## 1. Introduction

High-risk (HR) human papillomavirus (HPV) persistent infection has been widely recognized as the main causal risk factor for the development of cervical intraepithelial neoplasia (CIN) and progression to cervical cancer [1,2].

Nowadays, an increasing number of HPV tests, which differ for technology, targets, and genotyping [3], is available for HPV detection. Unfortunately, few of these have been studied and even less validated for screening [4].

The threshold of validated HR-HPV tests is CIN2+ detection because this is the clinical target for screening. However, HPV tests have been employed not only for screening, but also as a triage test and test-of-cure for follow-up of women treated for precancerous lesions.

If many studies are not present in previous literature regarding HPV tests in the screening setting, even less is known about their use in the follow-up after surgical treatment [5]. The objectives of post-treatment follow-up testing are to confirm that treatment was effective, to identify recurrence early, and to reassure women. Therefore, looking for the persistence of the same HPV genotype identified at baseline would be helpful for stratifying the risk of CIN recurrence, also known as “treatment failure” [6,7].

The aim of the present study is to evaluate and to compare the performance of six HPV DNA tests and one HPV mRNA test from liquid-based cervical cytology samples, for the detection of CIN2+ at baseline and as “test-of-cure” during post-treatment follow-up. The secondary objective of the study is to determine the sensitivity and the specificity of different HPV tests in the settings of screening and post-treatment follow-up.

## 2. Patients and Methods

### 2.1. Population

All women aged between 25–61 years, scheduled to be conservatively treated for CIN2+ at the European Institute of Oncology (IEO), Milan, from January 2011 to June 2015, were enrolled. “Conservative treatment” included excisional procedures, such as Loop Electro-Excision Procedure (LEEP) and laser conization, and ablative procedures, such as laser vaporization, in cases of ectocervical lesion and no evidence of ICC at pre-treatment colposcopic-guided biopsies. The study was approved by the Institutional Ethical Committee (IEO S544 study), and informed consent was obtained from all women at enrollment. A ThinPrep PreservCyt (Hologic, Inc. Bedford, MA, USA) cervical sample was collected in all patients before treatment and at the first follow-up visit planned at 6 ± 3 months after surgical treatment, in order to perform cytology, Qiagen Hybrid Capture 2 (HC2) and Roche Linear Array HPV Test (Linear Array). Roche Cobas^®^ 4800 HPV Test (Cobas), Abbott RealTime High Risk HPV (RT), BD Onclarity HPV Assay (Onclarity), and Seegene Anyplex II HPV HR (Anyplex) test were carried out on a left-over aliquot. Hologic APTIMA mRNA assay (Aptima) has been performed placing an aliquot of Thin Prep sample in the Aptima storage liquid upon arrival in the laboratory. The results of histology at baseline and at relapse, when occurred, were available for all patients. Principal characteristics of all HPV tests are detailed in Table 1.

### 2.2. Hybrid Capture 2

Qiagen HC2 test is a sandwich capture molecular hybridization assay: it is a signal amplification detection method based on chemiluminescence that detects 13 HR HPV types: HPV 16, 18, 31, 33, 35, 39, 45, 51, 52, 56, 58, 59, and 68. The DNA:RNA hybrids are captured on a microplate, and the emitted light is measured in a luminometer as relative light units (RLU). Samples were considered as positive if the ratio RLU/cut-off was >1.0 (equivalent to 5000 copies/reaction). All samples with RLU between 1 and 2.5 should be retested, as requested in package insert instructions.

### 2.3. Linear Array

The Roche Diagnostics Linear Array test uses biotinylated PGMY09/11 consensus primers to amplify a 450-bp region of the L1 gene. It is capable of detecting 37 HPV genotypes: HPV6, 11, 16, 18, 26, 31, 33, 35, 39, 40, 42, 45, 51, 52, 53, 54, 55, 56, 58, 59, 61, 62, 64, 66, 67, 68, 69, 70, 71, 72, 73 (MM9), 81, 82 (MM4), 83 (MM7), 84 (MM8), IS39 e CP6108. The denatured PCR products were then hybridized to an array strip containing immobilized oligonucleotide probes. The results were visually interpreted by using the provided reference guide according to manufacturer’s protocol by two independent operators, and the results were compared to reach the final one.

### 2.4. Cobas 4800 HPV Test

Cobas is a real-time PCR-based test able to detect HR-HPV genotypes: HPV 16 and 18 are reported as single genotypes, as well as a group of 12 other HR-HPV types (31, 33, 35, 39, 45, 51, 52, 56, 58, 59, 66, and 68) are reported as HR positive readout. This fully automated test detects the same genotypes of HC2, which have been classified as HR by (the International Agency for Research on Cancer (IARC)), and in addition HPV66, and includes an internal control (B-globin) as the marker of sample adequacy.

### 2.5. Real Time HR HPV

The Abbott RealTime HR HPV test (Abbott, Wiesbaden, Germany) is a qualitative real-time PCR for the detection of DNA from 12 HR-HPV genotypes (16, 18, 31, 33, 35, 39, 45, 51, 52, 56, 58, 59, 66, and 68), and includes an internal control (B-globin) as the marker of sample adequacy. Even an RT test is able to provide the identified genotype: for HPV 16 and 18 as single and for other HR genotypes in the pool, respectively.

### 2.6. Onclarity

The BD Onclarity HPV Assay detects 14 HR-HPV genotypes and co-amplifies a beta-globin internal control (IC) that acts as processing control. The primers for the 14 HPV genotypes are designed to target a region of 79–137 bases in the E6/E7 genome, whereas the IC primers amplify a 75 base region in the human beta-globin gene. The assay consists of three PCR assay tubes (G1, G2, and G3) and four optical channels for the detection of the 14 HR-HPV genotypes (16, 18, 31, 45, 51, 52) as single infections and the remaining eight genotypes in three groups (33/58, 56/59/66, 35/39/68) and the IC.

### 2.7. Anyplex II

The Seegene Anyplex II HPV HR test simultaneously detects 14 HR-HPV genotypes (HPV 16, 18, 31, 33, 35, 39, 45, 51, 52, 56, 58, 59, 66, and 68), and co-amplifies a beta-globin internal control (IC), which acts as processing control in only one real-time PCR reaction based on TOCE (tagging oligonucleotide cleavage and extension) technology. In case of positivity, Anyplex also provides the information of semi-quantitative viral load level of amplification, which can be measured repeatedly at 30, 40, and 50 cycles during the PCR process.

### 2.8. Aptima

The APTIMA HPV Assay searches for E6/E7 HR-HPV mRNA by three main steps, which take place in a single tube: target capture, target amplification through amplification mediated by the transcription (Transcription-Mediated Amplification, or TMA), and detection of amplification products (amplicons) by Hybridization Protection dosage Assay. The assay incorporates an internal control for the capture, amplification, and detection of nucleic acid, as well as any operator or instrumentation errors.

### 2.9. Cytology

The physician-collected ThinPrep PreservCyt cervical specimens were processed in the ThinPrep 2000 machine (Cytyc Corporation, Boxborough, Mass). All liquid-based cytology slides were stained according to the Pap method and all cytologic diagnosis were performed by trained specialist biotechnicians, following automated Focal Point evaluation of all slides. In case of abnormal cytology, a dedicated pathologist from the Cytology Unit of IEO reviewed cytology slides to confirm diagnosis. Results were reported according to the 2001 Bethesda Reporting System.

### 2.10. Histology

All histological diagnoses were made on a colposcopic-guided biopsy of the transformation zone alone or with endocervical curettage or on excision surgical specimens, by dedicated gynecological pathologists working at the Division of Pathology of IEO.

### 2.11. Statistical Methods

Patients’ characteristics were summarized by count and percent or mean and standard deviation (SD) for categorical and continuous variables, respectively. HPV test agreement, at both baseline and at follow-up, were estimated by the proportion of concordant cases. In order to take into account the low prevalence of negative and positive cases at baseline and at follow-up, respectively, concordance was estimated by the prevalence-adjusted and bias-adjusted kappa (PABAK) [14] statistic. Point estimates were tabulated alongside 95% confidence interval and the significance of the agreement between each HPV test with the HC2 test was determined by using the McNemar test. Sensitivity and specificity of each test at baseline and at follow-up were plotted in a forest-like plot for all patients and tabulated for the CIN2+ patients only. All tests were two-tailed and considered significant at the 5% level. The HC2 test was used as the reference test. All analyses were conducted using SAS 9.4 (N.C, Cary) and STATA (StataCorp. 2021. *Stata Statistical Software: Release 17*. College Station, TX, USA: StataCorp LLC).

## 3. Results

One hundred and seventy-two women scheduled to be conservatively treated with LEEP or laser conization or laser vaporization for CIN2+ were enrolled. The main characteristics of the study population at baseline are listed in Table 2.

Not all HPV tests were performed at baseline and follow-up due to lack of supply of reagents. Histological diagnosis on surgical specimens at baseline confirmed a CIN2+ lesion in 158 (91.9%) patients. Only histology confirmed samples were included in the final analysis.

Overall, 150 patients were recruited at post-treatment follow-up, but only 118 in a time range between 3 and 6 months. Twenty-two patients (12.8%) were lost to follow-up and 32 (21.3%) were excluded due to incorrect timing of test-of-cure. Assuming HC2 as comparator, all HPV tests employed showed a good degree of comparison at both baseline (PABAK ranging from 0.81 to 0.90) and follow-up (PABAK ranging from 0.53 to 1). The concordance between different HPV tests and HC2 ranges from 91% to 95% at baseline and from 76% to 100% at follow-up, respectively, as shown in Table 3 and Table 4.

Sensitivity and specificity of all employed tests for CIN2+ at baseline and at follow-up, compared to HC2, are summarized in Figure 1 and Figure 2. All HPV tests showed a very good sensitivity in detecting CIN2+ at baseline, more than 92%, and a very good specificity at follow-up, more than 89%.

## 4. Discussion

The results of our study showed a very good concordance among different HPV tests performed in liquid-based cervical samples from a group of women with high prevalence of preneoplastic cervical disease. The confidence intervals of these concordances overlap, further demonstrating the similarities of these HPV tests in performance. These data are in agreement with previous studies summarized in the 2020 list of human papillomavirus assays suitable for primary cervical cancer screening, published by Arbyn et al. (Arbyn et al., 2021). All HPV tests employed at baseline and follow-up have been validated according to Meijer’s guidelines [9,10,11,12,13]. As requested by validation guidelines (Meijer’s guidelines or Valgent protocol), the relative sensitivity and specificity must be high to be considered “validated”. Only LA is a test not fully validated according to Meijer’s guidelines, due to the additional search for low-risk (LR) HPV genotypes and the high sensitivity that does not correlate with CIN2+. However, data regarding positivity for LR-HPV have not been included in our analysis. Moreover, LA is a test previously validated according to Valgent protocol [8].

Since validations have been usually performed in the screening setting, these data are only indicative for baseline. In the present analysis, we focused on comparing tests’ performance not only at baseline, but also at the post-treatment follow-up. Interestingly, our data showed comparable performance of the tests in terms of sensitivity and specificity at both baseline and test-of-cure.

Due to the setting of samples, which show a high prevalence of positive at baseline and negative at follow-up, respectively, sensitivities were found to be higher at baseline and lower at follow-up. On the contrary, specificities are notably higher at follow-up than at baseline. Due to the low prevalence of HPV after treatment, we chose the prevalence and bias adjusted kappa, instead of either the simple or the weighted kappa, to estimate the HPV tests’ agreement.

However, all tests perform similarly at baseline and follow-up. Although Cobas seems to show better performance than other tests, these data might suffer from a bias related to the smaller number of samples that have been tested with the Cobas method.

A negative HPV result at follow-up provides a good negative predictive value. Indeed, we found only a case of disease recurrence in the cohort of patients with a post-treatment negative HPV test result, for any validated HPV test. In this patient, cytology was HSIL at follow-up and only LA revealed the presence of HPV18 and 73 at baseline, with the persistence of HPV 73 at relapse. Actually, LR HPV genotypes are not detected by validated HPV tests.

Furthermore, HPV tests that provide partial or extended genotyping showed comparable results.

The Aptima test, which detects HPV mRNA, showed no particular advantages in terms of sensitivity or specificity: the test performances are in line with other tests that detect HPV DNA.

Strengths of the present study include the type of population (women with only confirmed CIN2 + histology) and the timing of test-of-cure that was performed at 6 ± 3 months, as also suggested by the most recent guidelines from the Italian Group for Cervical Cancer Screening (GISCi) [15]. On the contrary, the main limit consists in the impossibility of performing all HPV tests in all samples.

In conclusion, our results demonstrate that validated HPV tests produce comparable results, and this cannot be extended to non-validated tests without proven evidence. Thus, only the use of validated HPV DNA or RNA tests is strongly recommended in both screening and test-of-cure settings. Moreover, HPV genotyping could be helpful in post-treatment management, by identifying women at higher risk of CIN2+ recurrence, due to the persistence of the same HPV genotype, and reassuring women who may present new HPV genotype infection after surgical treatment.

## Figures and Tables

**Figure 1 diagnostics-12-02359-f001:**
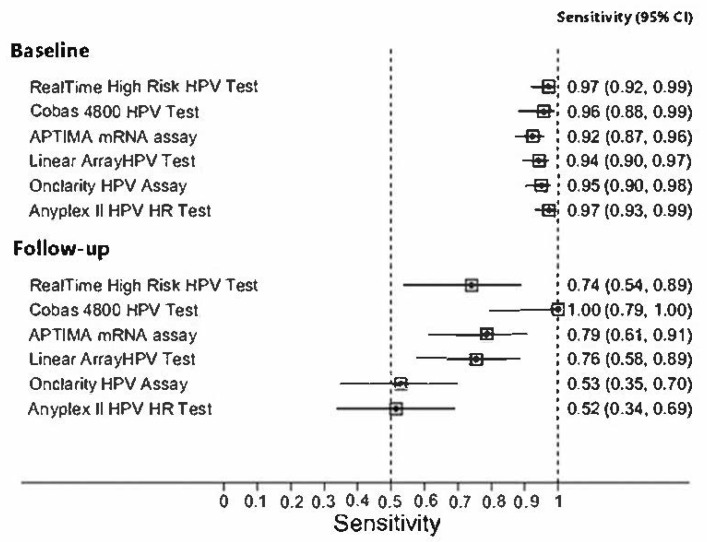
Sensitivity for CIN2+ at baseline and at follow-up.

**Figure 2 diagnostics-12-02359-f002:**
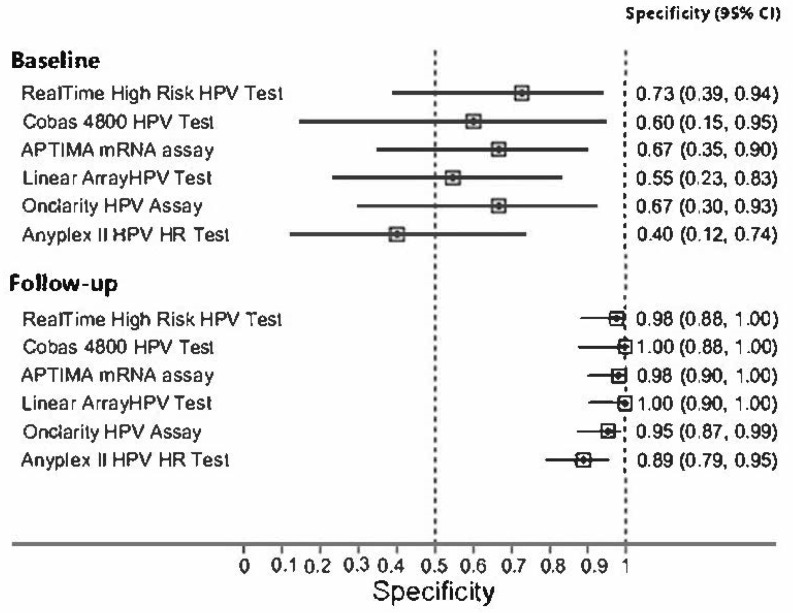
Specificity of all employed tests for CIN2+ at baseline and at follow-up.

**Table 1 diagnostics-12-02359-t001:** HPV test features.

Test	Company	Method	HPV TARGET	TARGET Region	HR HPV Genotypes														Validation References
					16	18	31	33	35	39	45	51	52	56	58	59	66	68	
Hybrid Capture II	Qiagen	Hybridization and signal amplification	DNA	Whole genome	●	●	●	●	●	●	●	●	●	●	●	●		●	Gold standard, NTCC study
Linear Array HPV Test	Roche	PCR and oligonucleotide hybridization	DNA	L1	●	●	●	●	●	●	●	●	●	●	●	●	●	●	[8]
Cobas 4800 HPV Test	Roche	Real Time PCR	DNA	L1	●	●	●	●	●	●	●	●	●	●	●	●	●	●	[9], ATHENA study
RealTime High Risk HPV Test	Abbott	Real Time PCR	DNA	L1	●	●	●	●	●	●	●	●	●	●	●	●	●	●	[10]
Onclarity HPV Assay	BD	Real Time PCR	DNA	E6 E7	●	●	●	●	●	●	●	●	●	●	●	●	●	●	[11]
Anyplex II HPV HR Test	Seegene	TOCE Real Time PCR	DNA	L1	●	●	●	●	●	●	●	●	●	●	●	●	●	●	[12]
APTIMA mRNA assay	Hologic	Transcription-Mediated Amplification	mRNA	E6 E7	●	●	●	●	●	●	●	●	●	●	●	●	●	●	[13]
in pool																			
in small pool																			

**Table 2 diagnostics-12-02359-t002:** Patients’ Characteristics at Baseline.

Characteristic	Level	Statistic *^a^*
Age, years		39.0 (7.8) *^b^*
Histology	CIN 1	14 (8.1)
	CIN 2+	158 (91.9)
HC2	Negative	12 (7.0)
	Positive	160 (93.0)
Cytology	Negative	6 (3.5)
	ASCUS	6 (3.5)
	LSIL	18 (10.5)
	HSIL/ASC-H	107 (62.2)
	AGC	2 (1.2)
	SCC	7 (4.1)
	*missing*	26 (15.1)

*^a^* Statistics are: Mean (SD) for Age, N (%) otherwise; SD = Standard Deviation; ***^b^*** min = 25, max = 61 years.

**Table 3 diagnostics-12-02359-t003:** HPV Test Results at Baseline Compared to Hc2 (Reference).

	HC2, N (col %) *^a^*					
HPV Test		Negative	Positive	PABAK (95% CI)	*p*-Value *^b^*	Agreement % (95% CI)
Abbott	Negative	8 (72.7)	3 (2.9)	0.90		95%
	Positive	3 (27.3)	100 (97.1)	(0.81, 0.98)	1.00	(88.9, 98.0)
Roche	Negative	3 (60.0)	3 (4.3)	0.87		93%
	Positive	2 (40.0)	67 (95.7)	(0.75, 0.98)	1.00	(85.1, 97.8)
Aptima	Negative	8 (66.7)	12 (7.6)	0.81		91%
	Positive	4 (33.3)	147 (92.5)	(0.73, 0.90)	0.08	(85.3, 94.6)
Linear Array	Negative	6 (54.6)	9 (5.7)	0.84		92%
	Positive	5 (45.4)	150 (94.3)	(0.75, 0.91)	0.42	(86.6, 95.4)
BD Onclarity	Negative	6 (66.7)	7 (4.9)	0.87		93%
	Positive	3 (33.3)	136 (95.1)	(0.79, 0.95)	0.34	(88.2, 96.8)
Seegene	Negative	4 (40.0)	4 (2.8)	0.87		93%
	Positive	6 (60.0)	137 (97.2)	(0.79, 0.95)	0.75	(88.2, 96.8)

***^a^*** Column percent on non-missing counts; ***^b^*** McNemar test exact *p*-Values. PABAK = Prevalence and Bias adjusted Kappa; 95%CI = 95% Confidence Interval.

**Table 4 diagnostics-12-02359-t004:** HPV Test Results at Follow-Up *^a^* Compared to Hc2 (Reference).

	HC2, N (col %) *^b^*					
HPV Test		Negative	Positive	PABAK (95% CI)	*p*-Value *^c^*	Agreement % (95% CI)
Abbott	Negative	43 (97.7)	7 (25.9)	0.78		89%
	Positive	1 (2.3)	20 (74.1)	(0.63, 0.92)	0.07	(79.0, 95.0)
Roche	Negative	28 (100)	0	1.00		100%
	Positive	0	16 (100)	(1.00, 1.00)	1.00	(92.0, 100)
Aptima	Negative	52 (98.1)	7 (21.2)	0.81		88%
	Positive	1 (1.9)	26 (78.8)	(0.69, 0.94)	0.07	(80.1, 93.1)
Linear Array	Negative	36 (100)	8 (24.2)	0.77		88%
	Positive	0	25 (75.8)	(0.62, 0.92)	0.008	(78.4, 94.9)
BD Onclarity	Negative	63 (95.5)	16 (47.1)	0.62		81%
	Positive	3 (4.6)	18 (52.9)	(0.47, 0.77)	0.004	(71.9, 88.2)
Seegene	Negative	57 (89.1)	16 (48.5)	0.53		76%
	Positive	7 (10.9)	17 (51.5)	(0.36, 0.70)	0.09	(66.6, 84.3)

***^a^*** N = 118 patients with first visit at 3–6 months, median f.u. days = 108, (min = 90, max = 179) ***^b^*** Column percent on non-missing counts; ***^c^*** McNemar test; PABAK = Prevalence and Bias adjusted Kappa; 95%CI = 95% Confidence Interval.

## Data Availability

The data presented in this study are available on request from the corresponding author. The data are not publicly available due to patients’ privacy restrictions. The data are safely stored in a private database of the European Institute of Oncology, Milan, Italy.

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
