# Peer review of "HPV Tests Comparison in the Detection and Follow-Up after Surgical Treatment of CIN2+ Lesions"

_diagnostics, 2022, doi:10.3390/diagnostics12102359_

Round 1

Reviewer 1 Report

the work provides an important analysis, especially at a time when attention in developed countries is focused on the WHO recommendation for the eradication of cervical tumors causally associated with high-risk Human Papillomavirus infection where the tests to be used for the screening of these infections will be of great importance in the early detection of patients at risk for the development of these tumors.

Author Response

We thank the reviewer for his efforts in reviewing our manuscript and his comments. 

Reviewer 2 Report

Please check that all references are relevant to the contents of the
manuscript.

Author Response

We thank the reviewer for his efforts in reviewing our manuscript and his comments. English language has been re-checked. We considered all references as relevant for the paper.

Reviewer 3 Report

Interesting study, especially the part concerning the post treatment results.

I have a few suggestions:

In patients and Methods.

Population. In the first row - conservative treatment/management of CIN2+ would be follow up/surveillance, and you mention afterwards that patients received surgical treatment (I suppose LLETZ/LEEP or cold knife conization). This should be cleared, and explained in text. This should be also changed/corrected in the first row of Results.

Histology. If you had final surgical histology diagnosis (on excision surgical specimens) it should be stated here together with biopsy.

Results. Sensitivity and specificity of cytology should be mentioned, no matter it was not a part of the study aims.

Discussion. It will be good if you would elaborate more and give the opinion about the possible reasons of such unexpected results of low sensitivity and high specificity of HPV tests in Follow up.

Author Response

We thank the Reviewer for his efforts in reviewing our manuscript and his suggestions.

Please find below our point-by-point replies:

- Definition of conservative treatment was added in the Material and Methods section (lines 87-89) and in Result section (lines 183-184) as suggested.

- Excision surgical specimens has been stated in the Methods section as requested (line 164).

- Sensitivity and specificity of cytology has been calculated for CIN2+ detection at baseline and were 0.86 (95%CI: 0.80-0.91) and 0.17 (95%0CI: 0.02-0.48), respectively. Conversely, at follow-up cytology showed a sensitivity of 0.24 (95%CI: 0.15-0.36) and a specificity of 0.95 (95%CI: 0.89-0.98). We decided not to include these results in the manuscript because our analysis is focused on comparing HPV test performances at baseline and at post-treatment follow-up. Moreover, it is already well known that HPV tests are superior to cytology in terms of sensitivity.

- The unexpected observed low sensitivity and high specificity of HPV tests in follow-up are both related to the low prevalence of HPV after treatment. This is also the reason why we chose the Prevalence and Bias Adjusted Kappa, instead of either the simple or the weighted kappa, to estimate the HPV tests agreement. We better specified this reason in the Discussion section as suggested (lines 255-257).

In the end, English language and style has been re-checked.